# MRI- and Histologic-Molecular-Based Radio-Genomics Nomogram for Preoperative Assessment of Risk Classes in Endometrial Cancer

**DOI:** 10.3390/cancers14235881

**Published:** 2022-11-29

**Authors:** Veronica Celli, Michele Guerreri, Angelina Pernazza, Ilaria Cuccu, Innocenza Palaia, Federica Tomao, Violante Di Donato, Paola Pricolo, Giada Ercolani, Sandra Ciulla, Nicoletta Colombo, Martina Leopizzi, Valeria Di Maio, Eliodoro Faiella, Domiziana Santucci, Paolo Soda, Ermanno Cordelli, Giorgia Perniola, Benedetta Gui, Stefania Rizzo, Carlo Della Rocca, Giuseppe Petralia, Carlo Catalano, Lucia Manganaro

**Affiliations:** 1Department of Radiological, Oncological and Pathological Sciences, Policlinico Umberto I, Sapienza University of Rome, 00161 Rome, Italy; 2Department of Computer Science & Centre for Medical Image Computing, University College London, London WC1E 6BT, UK; 3AINOSTICS, Manchester M2 3NG, UK; 4Radiomics Core Research Facility, Fondazione Policlinico Universitario “A.Gemelli” IRCCS, 00168 Roma, Italy; 5Department of Medico-Surgical Sciences and Biotechnologies, Polo Pontino-Sapienza University, 04100 Latina, Italy; 6Department of Maternal and Child Health and Urological Sciences, Sapienza University of Rome, Policlinico Umberto I, Viale del Policlinico 166, 00161 Rome, Italy; 7Department of Radiology, European Institute of Oncology IRCCS, 20141 Milan, Italy; 8Division of Gynecologic Oncology, European Institute of Oncology, Istituto di Ricovero e Cura a Carattere Scientifico (IRCCS), 20099 Milano, Italy; 9Department of Medicine and Surgery, University of Milano-Bicocca, 20126 Monza, Italy; 10Medical Oncology Department, Università Campus Bio-Medico di Roma, 00128 Rome, Italy; 11Unit of Diagnostic Imaging, Università Campus Bio-Medico di Roma, 00128 Rome, Italy; 12Unit of Computer Systems and Bioinformatics, Department of Engineering, University Campus Bio-Medico di Roma, 00128 Rome, Italy; 13Dipartimento di Diagnostica per Immagini, Radioterapia Oncologica ed Ematologia, Fondazione Policlinico Universitario Agostino Gemelli, IRCCS, 00168 Rome, Italy; 14Clinica di Radiologia EOC, Istituto Imaging della Svizzera Italiana (IIMSI), 6900 Lugano, Switzerland; 15Precision Imaging and Research Unit, Department of Medical Imaging and Radiation Sciences, European Institute of Oncology IRCCS, 20141 Milan, Italy

**Keywords:** endometrial cancer, magnetic resonance imaging (MRI), texture analysis, radiomic analysis, radio-genomic analysis, lymph-vascular space invasion, risk classes, molecular classification

## Abstract

**Simple Summary:**

Our study showed the potential of whole tumor radio-genomic-based analysis for the preoperative evaluation of endometrial cancer (EC). Since radio-genomics can provide information regarding high-risk factors from standard preoperative MR images, radio-genomic-based models could be useful in preoperative risk stratification of EC patients and prediction of lymphatic-vascular infiltration (LVSI) before surgery. Predictive radio-genomics models based on T2WI and ADC texture features showed a medium-to-high diagnostic performance in predicting low-risk EC and LVSI. The innovative clinical impact of our investigation is to demonstrate how radio-genomic analysis can be supportive in the assessment of some parameters considered incomplete and inaccurate after a preoperative MRI and biopsy such as LVSI assessed only after post-surgical histologic analysis; myometrial infiltration, which is highly operator-dependent on MRI; and frequent discordance between preoperative and post-hysterectomy findings for tumor grade. Application of predictive models in clinical practice would lead to significant advantages in the preoperative selection of individualized therapy and reductions in time/cost.

**Abstract:**

High- and low-risk endometrial carcinoma (EC) differ in whether or not a lymphadenectomy is performed. We aimed to develop MRI-based radio-genomic models able to preoperatively assess lymph-vascular space invasion (LVSI) and discriminate between low- and high-risk EC according to the ESMO-ESGO-ESTRO 2020 guidelines, which include molecular risk classification proposed by “ProMisE”. This is a retrospective, multicentric study that included 64 women with EC who underwent 3T-MRI before a hysterectomy. Radiomics features were extracted from T2WI images and apparent diffusion coefficient maps (ADC) after manual segmentation of the gross tumor volume. We constructed a multiple logistic regression approach from the most relevant radiomic features to distinguish between low- and high-risk classes under the ESMO-ESGO-ESTRO 2020 guidelines. A similar approach was taken to assess LVSI. Model diagnostic performance was assessed via ROC curves, accuracy, sensitivity and specificity on training and test sets. The LVSI predictive model used a single feature from ADC as a predictor; the risk class model used two features as predictors from both ADC and T2WI. The low-risk predictive model showed an AUC of 0.74 with an accuracy, sensitivity, and specificity of 0.74, 0.76, 0.94; the LVSI model showed an AUC of 0.59 with an accuracy, sensitivity, and specificity of 0.60, 0.50, 0.61. MRI-based radio-genomic models are useful for preoperative EC risk stratification and may facilitate therapeutic management.

## 1. Introduction

Endometrial carcinoma (EC) is the most common gynecologic malignancy in the Western world and the fourth leading cause of cancer death in women [1]. It is a typical neoplasm of the post-menopausal age (75% are >50 years) with an average age at diagnosis of about 63 years [2,3,4]. EC is diagnosed at an early stage (Stage I) in 80% of cases because it causes atypical blood loss, a red flag in postmenopausal patients, and this correlates with a favorable outcome (5-year survival rates above 95%) [2]. Nevertheless, 15–30% of patients have a relapse of the disease [5]. Prognosis and probability of recurrence depend on the histologic type, grading, lymph-vascular space invasion (LVSI) and staging according to the International Federation of Gynecology and Obstetrics (FIGO 2018). Identifying preoperatively which patients are most at risk of recurrence may have a great impact on therapeutic management, avoiding over- and under-treatment and prolonging survival. The first examination performed in women with atypical vaginal bleeding when endometrial carcinoma is suspected is transvaginal ultrasound (TVS-US). TVS-US, despite its highly operator-dependent nature, allows accurate assessment of tumor extent and myometrial infiltration when conducted by experienced operators.

The diagnosis of EC requires histology and hysteroscopy with endometrial biopsy being generally used to assess histology subtype and grading preoperatively. Subsequently, preoperative EC staging is assessed by Magnetic Resonance Imaging (MRI), which represents the gold standard for preoperative locoregional evaluation according to FIGO 2018 classification. The standard treatment is bilateral hysterosalpingo-oophorectomy and sentinel lymph node (SLN) dissection, while lymphadenectomy and adjuvant therapy are performed only if required by the EC risk factor (Figure 1).

In order to provide therapeutic management guidelines, the ESMO-ESGO-ESTRO Consensus Conference 2016 stratified EC into four risk classes according to histology, grade, stage and the presence of LVSI, which should reflect tumor aggressiveness and the likelihood of recurrence directing toward possible adjuvant therapy [6].

Recent scientific evidence proved that patients belonging to the same ESMO-ESGO-ESTRO 2016 risk class have discordant disease progression and sometimes low-risk patients may show unexpectedly shorter progression-free survival (PFS) than high-risk patients [5]. This inconsistency highlighted the potential benefit of novel predictive tissue markers that can integrate preoperative risk classification.

Recent work from the Cancer Genome Atlas (TGCA) indicates that there are four genomic classes of endometrioid carcinoma, each with a different mutational profile, which can be used to stratify these tumors in distinctive epidemiologic, morphologic, prognostic and therapeutically predictive categories. The most prevalent group is the “copy number low” (CN-L), which are tumors with an absence of large numbers of genomic amplifications and deletions, microsatellite instability and POLE exonuclease domain hotspot mutations. The next prevalent group is the microsatellite instability-high (MSI-H), defined by the presence of MSI-H and “hypermutation” in the absence of POLE exonuclease domain hotspots mutations.

The next two groups contain comparable numbers of endometrioid carcinoma: POLE and CN-H (or “serous like”).

The POLE group has hotspot mutations in the exonuclease domain of POLE and is characterized by the presence of the highest numbers of mutations (expressed as mutations per megabase and referred to as “ultramutated”).

The serous-like group (CN-H) is defined by the presence of large numbers of genomic amplifications and deletions but a low number of mutations per megabase; nearly all cases have Tp53 mutations [7]. AFIP—Atlas of tumor pathology—Tumors of Uterine Corpus and Trophoblastic Disease, E. Oliva et al., 2020.

Inspired by these new prospectives, the Proactive Molecular Risk Classifier for Endometrial Cancer (“ProMisE”) has simplified the above-mentioned classification establishing four similar genomics groups represented by a POLE-mut, MMR-d, NMDR, and p53abn class, testing it on a large validating population [8,9,10]. The molecular preoperative assessment may be used not only for risk stratification but also for tailored surgical decisions, adjuvant therapy and individualized follow-ups [5]. For these reasons, the recent ESGO/ESTRO/ESP 2020 guidelines for the management of EC patients added this molecular stratification to the 2016 guideline, keeping the division into four risk classes, as shown in Table 1 [11].

However, not all the mentioned parameters can be assessed preoperatively. In particular, the presence of LVSI is provided only by post-hysterectomy pathological examination. Moreover, a poor agreement was observed between post-surgical histology and presurgical biopsy for tumor grade, probably due to the small amount of tissue examined with a biopsy [12,13]. Furthermore, post-surgical assessment of myometrial infiltration may not be consistent with the preoperative MRI evaluation. Therefore, the preoperative investigations do not allow for complete risk stratification according to the ESGO/ESTRO/ESP guidelines. This condition may lead to possible over- or under-treatment of patients; especially for low-risk patients who do not undergo lymphadenectomy and adjuvant treatment (Figure 1) [14,15]. Therefore, the proper application of this molecular risk stratification (ESGO/ESTRO/ESP 2020) is mainly obstructed by the inability to obtain all the required information preoperatively, as well as by the limited availability and high cost of molecular analysis. These conditions limit a systematic approach to the personalized therapeutic management of endometrial cancer.

MRI is considered the gold standard in preoperative assessment of EC but the radiologist’s eye cannot identify the different and subtle tissue compositions (i.e., “texture characteristics”) that, instead, can be assessed with computerized instruments. In this context, we hypothesize that the information contained in MRI-derived images can be leveraged to distinguish between high- and low-risk classification under the ESGO/ESTRO/ESP 2020 standard.

Radiomics is an innovative method to extract quantitative information about microscopic and mesoscopic characteristics of tissues from clinical images; this data can be combined with histology, molecular data or other information in order to build new models and novel biomarkers to predict specific targets, such as diagnosis, treatment response, survival or genomic and proteomic alterations [16].

Despite promising results, only a few studies have explored MRI-based radiomic tumor features in EC and preoperative aggressiveness assessment; to the best of our knowledge, this is the first radiomic study approaching new molecular risk classification for EC [17,18,19]. We aim to develop and validate an MRI-based radiomic model that may preoperatively predict the EC low-risk class according to the recent ESGO/ESTRO/ESP 2020 molecular risk classification. Therefore, we aim to preoperatively identify which patients will not need to undergo lymphadenectomy and adjuvant treatments, establishing a tailored treatment to the patient’s risk class.

## 2. Materials and Methods

### 2.1. Patient Population

This is a multicentric, retrospective study conducted from October 2015 to August 2022. In our study, patients with a first diagnosis of EC were enrolled in two different centers: Center 1 was the Umberto I Hospital, Sapienza University of Rome (Group 1), Center 2 was the European Institute of Oncology (IEO), Milan (Group 2).

The enrolled patients were retrospectively identified from the database of the department of gynecological oncology and from Picture Archiving and Communication System (PACS) of radiology department to fulfill the inclusion criteria:○Subject age >/= 18 years;○Available written informed consent;○Histologic diagnosis of EC;○Preoperative 3 T MRI images;○Surgery for staging or complete debulking.

Exclusion criteria were:○MR images of poor diagnostic quality due to presence of artifacts;○Spoiled histology due to fixation artifacts;○Neoadjuvant chemotherapy or radiotherapy implemented before MRI examination.

A total of 130 patients were first enrolled, subsequently, 66 patients were excluded as illustrated in the flowchart of Figure 1. The final study population consisted of 64 women with a mean age of 66 ± 11.5 [SD] years (range: 45–88 years): 49 enrolled in Group 1 and 15 enrolled in Group 2.

For the predictive risk classification model, patients were first stratified according to the ESGO/ESMO/ESP 2020 molecular classification and then divided into two groups: low-risk (including patients belonging to the low-risk class according to the 2020 molecular classification) and high-risk (including intermediate, intermediate–high and high-risk classes).

### 2.2. Gynecological Procedure

All the included patients underwent surgery for staging or complete debulking with histologic diagnosis of EC. Surgical techniques included laparoscopic or laparoscopic robotic assisted with Da Vinci X System approaches. With these minimally invasive techniques, access to the abdominal cavity was performed with umbilical trocar using open technique and subsequently, pneumoperitoneum of about 12 mmHg was induced. In the case of laparoscopic surgery, under direct visualization, three 5 mm ancillary trocars were inserted in the abdomen (one suprapubic and two laterally to the epigastric arteries, in the left and right lower abdominal quadrants, respectively). In the case of robotic surgery, in addition to the umbilical trocar, two 8 mm robotic trocars were used in left and right subcostal areas. In addition, a 12 mm accessory trocar was placed between the umbilicus and the left robotic trocar for the assistant. All patients underwent lymph nodal staging through bilateral sentinel lymph node (SLN) mapping. To delineate the courses of lymphatic vessels, cervical injection of tracer indocyanine green fluorescent was provided, under direct visualization. Laparoscopic inspection of abdominal cavity was performed using a fluorescence imaging system. After the access in the retroperitoneum, sentinel nodes were identified and removed. Finally, extrafascial type A extra-fascial hysterectomy and bilateral salpingo-oophorectomy were performed.

Clinical parameters, including patients’ age, date of surgery, family history of cancer, hypertension, diabetes, Body Mass Index (BMI), endocrinopathy, adenomyosis and stage according to FIGO 2018 criteria were collected from clinical charts.

### 2.3. Pathological Analysis

From the database of pathology department, we retrieved samples of endometrial carcinoma (EC) in order to complete immunohistochemical stains requested to assess molecular classification of neoplasm: POLE-mutant EC, MMR (mismatch repair)-deficient EC, p53 mutant EC, NSMP (no specific molecular profile)-EC, (sec. WHO classification of Female Genital Tumours, 2019) (Figure 2).

For each tumor sample, hematoxylin and eosin-stained slides were blindly reviewed by two pathologists; based on the dimension of each tumor, a variable number of slides (from three to six) were available. A complete histological assessment of each lesion was performed, which included pattern of growth, stromal features such as desmoplasia, inflammation, lymphoid hyperplasia and/or necrosis, tumor grading and finally the presence of lymph vascular invasion.

We refer to ProMisE algorithm, so the first assessment is immunohistochemistry (IHC) for the presence of mismatch repair proteins PMS2, MLH1, MSH2, MSH6, with loss/deficient tumors categorized as MMR-D. Next, tumors are assessed for mutations in polymerase-E exonuclease domain, exons 9–14 (POLE), and finally for aberrant expression of p53 by IHC.

Immunohistochemistry was performed on paraffin slides representatively for each tumor with the Leica Bond 3 auto Stainer, using the primary antibodies to p53 (DO-1, mouse monoclonal), PMS2 (MORG4, mouse monoclonal), MLH1 (ES05, mouse monoclonal), MSH6 (PU29, mouse monoclonal) and MSH2 (79H11, mouse monoclonal) and estrogen (6F11, Novocastra) and progesterone (321, Novocastra) receptor, all purchased by Leica Biosystems, (Wetzlar, Germany). The signal was obtained with Bond Polymer Refine detection that contains peroxide block, post-primary, polymer reagent, DAB chromogen (brown signal) and Hematoxylin counterstain. The section was dehydrated and mounted.

#### Molecular Analysis for Detection of POLE Mutation

Molecular analysis was performed on formalin-fixed and paraffin-embedded (FFPE) samples with a minimum tumor cellularity of 10%. Tumor DNA was extracted using RecoverAll™ Total Nucleic Acid Isolation Kit for FFPE (ThermoFisher Scientific, Waltham, MA, USA). Extracted DNA samples were quantified and their fragmentation profiles were evaluated by real-time PCR analysis. Samples were amplified in two steps following the manufacturer’s instructions: PCR1 was performed to amplify hot-spot regions and PCR2 was performed to provide the indexed fragments.

Moreover, DNA was quantified on the EasyPGX qPCR platform with the EasyPGX Analysis Software V.4.0.10 (Diatech Pharmacogenetics, Jesi, AN, Italy). Finally, libraries were diluted at 50 pM and pooled together for template generation. NGS analysis was performed on an iSeq100 sequencer (Illumina, San Diego, CA, USA) according to the manufacturer’s instructions using the in vitro diagnostic (IVD) NGS panel ‘Myriapod NGS Cancer panel DNA’ (Diatech Pharmacogenetics, Jesi, Italy).

Data analysis for coverage and variant calling inspection was carried out by Myriapod NGS Data analysis Software V.5.0.4 (Diatech Pharmacogenetics, Jesi, AN, Italy). Samples with minimal coverage of 5000× and a variant alteration of ≥1% were selected.

### 2.4. MRI Protocol

In both centers, all MRI were acquired on 3T MR scanners (Umberto I Hospital: GE Discovery MR 750, GE Healthcare, Milwaukee, WI, USA and IEO Hospital: Magnetom Skyra, Siemens, Erlangen, Germany) using an eight-element pelvic phased-array surface coil. Patients were asked to fill their bladders halfway. The feet-first approach in the supine position was used. The standardized protocol included (Table 2): T2 Fast-Spin Echo (FSE) Weighted Imaging (WI) on sagittal, axial and coronal plane; T2 FSE WI oriented on the short axis (para-axial plane) and long axis (para-coronal plane) of the endometrial cavity; Field of View Optimized and Constrained Undistorted Single Shot (FOCUS) DWI on para-axial plane or sagittal with b-values of 0–500–1000 s/mm^2^ to obtain ADC maps.

For each patient we retrospectively extracted DICOM files of T2WI, DWI and ADC of preoperative pelvic MRI acquired on the same plane, mostly paraxial plane and rarely on sagittal plane. The extracted DICOM were screened for quality assurance by an expert radiologist to be selected for the radiomic analysis; MR images of poor diagnostic quality due to presence of artifacts were excluded.

### 2.5. Texture Analysis

#### 2.5.1. Tumor Segmentation

The endometrial primary tumors were manually delineated using the free, open-source software application ITKSNAP (www.itksnap.org, v. 3.8.0, accessed on 20 August 2022). T2WI, DWI and ADC DICOM files were converted in NIFTI and opened in ITKSNAP.

The whole tumor segmentations were performed by a radiologist (V.C.) with 4 years of expertise in gynecological imaging who manually outlined the region of interest (ROI) on the T2WI and DWI along tumor boundary on each slice; the segmentation was subsequently reviewed by senior radiologist with 25 years experience in pelvic female tumors (L.M.) (Figure 3). The radiologists were blind to the clinical and pathologic outcomes. The segmented tumor area appeared as intermediate/low intensity compared to the normal myometrium for T2WI, while it appeared as high intensity for DWI.

#### 2.5.2. Radiomic Features Extraction

Due to the limited number of subjects available in this study, to narrow down the number of extracted features, we focused our attention on T2WI and ADC maps, forgoing the DWI. For each subject, the T2WI and the ADC maps, together with the corresponding masks, were fed into Pyradiomics framework (https://pyradiomics.readthedocs.io/en/latest/, accessed on 4 September 2022) for the extraction of the radiomic features. All 8 available classes of features were selected, for a total of 120 features for each imaging modality. For each feature, boxplots of the values were produced to assess the presence of outliers or unexpected values.

### 2.6. Statistic

#### 2.6.1. Risk Classification Analysis

##### 2.6.1.1. Radiomic Features Selection

To ensure the robustness of the predictive model, a two-step feature selection was implemented. These steps were carried out using all the available data (Group 1 and Group 2). First, we assessed what features could individually separate the low from the high-risk groups via a non-parametric Wilcoxon–Mann–Whitney test. We controlled for multiple comparison false discovery rates following the Benjamini–Hochberg procedure. The second step of feature selection was implemented to eliminate highly correlated features. The correlation coefficient R was calculated pairwise from the set of statistically significant features; a pair of features was considered highly correlated for a |R| ≥ 0.8.

##### 2.6.1.2. Model Selection

We used a multiple logistic regression approach to classify between low- and high-class risk. Once identified the statistically significant features we had to choose a model from the set of candidates obtainable by combining all or a subset of these features. Due to the limited number of subjects available in this study we opted for models with a number of predictor variables at most equal to five. To select the most appropriate model, we adopted a probabilistic model selection approach that attempts to quantify both the model performance and model complexity. Specifically, we used an Akaike Information Criterion approach (AIC). We computed AIC for all the models with predictors obtained from considering the permutations of the 7 significant features into 1, 2, 3, 4 or 5 slots. The AIC was obtained considering both the number of predictors of each model and the loglikelihood of fitting each model to the entire set of data (Group 1 and Group 2).

##### 2.6.1.3. Model Training and Testing

Due to the limited number of subjects and the considerable class imbalance of Group 2 (high risk = 11, low risk = 4) we opted for 3-fold cross-validation approach to train and validate the selected model. The three sets were obtained by randomly allocating each of the subjects into one of the sets. The three sets were composed of a training set of 44 (31 Group 1, 13 Group 2), 42 (36 Group 1, 6 Group 2), and 44 (31 Group 1, 13 Group 2) subjects, respectively. The corresponding labels were 33/10 (high/low), 27/15 (high/low), 30/13 (high/low) for training and 12/9 (high/low), 18/4 (high/low), 15/6 (high/low) for testing. We iteratively trained on two sets and tested on the remaining one. For each iteration, we assessed the quality of the prediction in terms of area under the curve (AUC), accuracy, specificity and sensitivity.

#### 2.6.2. Model Training and Testing for LVSI

In addition to assessing the feasibility of predicting the risk class from T2w and DWI MRI-derived images, we also assessed whether it is possible to predict the tumor uterine infiltration (LVSI) from the same data.

The radiomic feature selection and model selection steps were similar to those described in the previous Section 2.6.1.1 and Section 2.6.1.2. However, in this framework, all the steps were performed considering only data from Group 1. The model was then trained on Group 1 data and tested on Group 2 as in this case the labels were more evenly distributed (Group 1: infiltration/no infiltration 20/29; Group 2: infiltration/no infiltration 6/9). Performance of the prediction was assessed via AUC, accuracy, specificity and sensitivity.

## 3. Results

### 3.1. Population

Patients’ characteristics, histological subtype, grade, and FIGO stage are presented in Table 3 and Table 4.

From the 49 patients selected from Center 1, the postoperative histologic assessment revealed myometrial invasion < 50% in 21 patients (42.85%), myometrial invasion ≥ 50% in 28 patients (57.15%); LVSI status revealed LVSI negative in 26 patients (50.06%) and LVSI positive in 23 patients (49.93%). The distribution of the molecular subtypes was as follows: 1/49 (2.04%) POLE, 19/49 (38.7%) MMR-D, 24/49 (48.97%) NSMP/p53wt and 6/49 (12.24%) p53 aberrant. No patient had tumors that demonstrated more than one molecular feature. The distribution of risk classes was as follows: 16/49 (32.61%) low risk, 7/49 (14.40%) intermediate risk, 11/49 (22.44%) intermediate–high risk and 15/49 (30.65%) high risk; subsequently they were sub-stratified in 16 patients classified as low class (32.65%) and 33 patients (67.34%) classified intermediate–high class. All Group 1 information is summarized in Table 4.

Of the 15 patients selected from Center 2, the postoperative histologic assessment revealed myometrial invasion < 50% in five patients (33.3%), myometrial invasion ≥ 50% in 10 patients (66.6%); LVSI status revealed LVSI negative in nine patients (60%) and LVSI positive in six patients (40%). The distribution of the molecular subtypes was as follows: 2/15 (13.33%) *POLE*, 5/15 (33.33%) MMR-D, 4/15 (26.66%) NSMP/p53wt and 4/15 (26.66%) p53 aberrant. No patient had tumors that demonstrated more than one molecular feature. The distribution of risk class was as follows: 3/15 (20%) low risk, 3/15 (20%) intermediate risk, 1/15 (6.67%) intermediate–high risk and 8/15 (53.33%) high risk subsequently they were sub-stratified in three patients classified as low class (20%) and 12 patients intermediate–high risk (80%). All Group 2 information is summarized in Table 4.

### 3.2. Risk Classification

Figure 4 shows the class types of the features that were found to be statistically significant after the Wilcoxon–Mann–Whitney test and Benjamini–Hochberg correction procedure. We found that 12 features were statistically significant with a corrected *p*-value p_(FDR)_ < 0.05 (T2w-derived = 5, ADC-derived = 7, Table 5). After removing those with high correlation (|R| ≥ 0.8), only seven features survived: Sum Entropy (GLCM) Gray Level Non-Uniformity (GLSZM), Small Area Low Gray Level Emphasis (GLSZM), Coarseness (NGTDM) from T2WI; Difference entropy (GLCM), Dependence entropy (GLDM), Dependence Variance (GLDM) from ADC.

We found that the best-performing model under the AIC was the model consisting of two features, one from T2w and one from ADC. The two features were: Coarseness (T2WI) and Difference Entropy (ADC).

Figure 5 shows the results obtained from the three-fold cross-validation approach. In each subplot, we report the training and test ROC curves obtained from training the best model on two subsets and testing on the remaining one. We found an AUC of 0.77, accuracy of 0.80, sensitivity of 0.82 and specificity of 0.5 for training set 1; an AUC of 0.78, accuracy of 0.76, sensitivity of 0.77 and specificity of 0.72 for training set 2; an AUC of 0.71, accuracy of 0.74, sensitivity of 0.75 and specificity of 0.66 for training set 3.

The averaged values for the training set were AUC 0.75, accuracy 0.76, sensitivity 0.78 and specificity 0.63. We found for test set 1 an AUC of 0.77, accuracy of 0.66, sensitivity of 0.63 and specificity of 1; for test set 2 an AUC of 0.65, accuracy of 0.72, sensitivity of 0.83 and specificity of 0.25; for test set 3 an AUC of 0.81, accuracy of 0.85, sensitivity of 0.83 and specificity of 1. The averages for the testing set were an AUC of 0.74, accuracy of 0.74, sensitivity of 0.76 and specificity of 0.94.

Table 6 shows the confusion matrix for each run, respectively, for the training and test sets. In order to assess whether each of the predictor variables has a statistically significant relationship with the response variable in the model, we estimated the *p*-values associated with the null hypothesis that the corresponding model parameter values obtained from fitting the model to each of the three different training sets were zero. We found that the values of the coefficients associated with both the Coarseness and Difference Entropy features had significant values two out of three times: p_run1_ < 0.05, p_run2_ > 0.05, p_run3_ < 0.05 for the Coarseness; p_run1_ < 0.05, p_run2_ < 0.05, p_run3_ > 0.05 for the Difference Entropy.

### 3.3. LVSI

Figure 6 shows the feature class types of the 82 features that were found to be significantly different (p_(FDR)_ < 0.05) between the group with LVSI and the group without it (T2w-derived = 40, ADC-derived = 42). The number of features decreased to 26 after removing those with high correlation.

We found that the best-performing model under the AIC was a model consisting of a single predictor feature from ADC: Large Dependence Emphasis (LDE). Figure 7 shows the ROC curve obtained from training the best model with Group 1 data and testing on Group 2 (AUC of 0.67 for training and an AUC of 0.59 when testing). We found an AUC of 0.67, accuracy of 0.65, sensitivity of 0.60 and specificity of 0.67 for the training set and for the testing set an AUC of 0.59, accuracy of 0.60, sensitivity of 0.50 and specificity of 0.61. We report the confusion matrices of both training and testing sets in Table 7. We found that the predictor variable and the intercept parameters of the model were significantly different from zero (p_predicto_ < 0.01, p_interc_ < 0.05).

## 4. Discussion

In this study, we showed the potential of whole tumor radiomic-based analysis for the preoperative evaluation of EC. Since radiomics can provide information regarding high-risk factors from standard preoperative MR images, radiomic-based models could be useful in the preoperative risk stratification of EC patients and the prediction of LVSI before surgery.

The predictive radio-genomic model based on T2W, ADC and DWI texture features showed a medium-to-high diagnostic performance in discriminating the class of interest: predict low-risk EC and LVSI. Our study demonstrated promising reproducibility and reliability, as confirmed by the external validation test and cross-validation test. The innovative clinical impact of our investigation is to demonstrate how radio-genomic analysis can be supportive in assessing preoperative risk factors; in particular, it could provide some parameters considered incomplete and inaccurate after preoperative MRI and biopsy.

Firstly, the radiologist’s eyes and biopsy specimens cannot identify LVSI, which assumes a key role in the risk stratification of EC. In fact, LVSI correlates with a higher frequency of tumor recurrence and lymph node metastasis, a reduced PFS and OS; therefore, its evaluation may lead toward adjuvant therapy even in EC that were preoperatively misclassified as early-stage tumors [20,21].

In recent years, some studies have related LVSI to texture features extracted from MRI images, obtaining satisfying predictive models for the assessment of LVSI. Yoshiko Ueno et al. developed a radiomic model based on T2-weighted, DWI and DCE images that provided high diagnostic performance in discriminating LVSI (AUC 0.80; sensitivity 80.9%; specificity 72.5%) [19]. The limitations of their study were the two-dimensional analysis of MRI images and correlation with only first-order textual features.

Another study, conducted by M. Bereby-Kahane, obtained a lower performance in LVSI prediction showing an AUC of 0.59, a sensitivity of 71% and a specificity of 59%; they performed a selection of six two-dimensional images to best sample the tumor lesion (two T2-weighted images and three ADC map images) [20].

The results of our predictive radio-genomic model for LVSI assessment were similar to those of M. Bereby-Kahanea’s study; our model was based on a single feature (Large Dependence Emphasis, GLSZM) extracted from the ADC map and showed an AUC of 0.67 for training and an AUC of 0.59 for the validation set. This selected feature quantifies tumor heterogeneity: thus a higher Large Dependence Emphasis (LDE) value is indicative of smaller dependence and more homogeneous texture; our results showed a higher LDE in LVSI-negative patients, supporting the hypothesis that more homogeneous tissue composition is compatible with a less aggressive tumor.

In agreement with our results, Ytre-Hauge et al. observed that high tumor entropy in ADC maps was correlated with high-risk tumors, specifically, they showed that high tumor entropy in ADC was correlated with high-risk histologic subtype (OR, 1.01; *p* = 0.004) and deep myometrial invasion (Odds ratio [OR], 3.2; *p* = 0.001) [22]. These findings were in agreement with those of Ganeshan B, et al., who observed that tumors with high intratumor heterogeneity have a poorer prognosis, potentially reflecting the tumor’s intrinsic aggressiveness [23]. Our radio-genomic model showed important differences from previous studies: first, we performed a whole-tumor MRI radiomic feature extraction, which provide more information about the real composition of the tumor; moreover, we validated the model on a different and external cohort of patients. The external validation probably also justifies the lower AUC value than that of Yoshiko Ueno et al. because our two groups (training and validation) were acquired in different centers, using different MRIs; this condition on the one hand may lead to a worse-performing model, but on the other hand, it assesses the generalizability and reproducibility of the radiomic profile of a whole-tumor MRI in the EC [20].

Moreover, our study has demonstrated that MRI-based whole-tumor texture analysis provides information regarding preoperative risk stratification of EC and is able to preoperatively predict low-risk class which required less extended surgical treatment.

High-risk patients according to non-molecular risk stratification of EC proposed by the ESMO-ESGO-ESTRO Consensus Conference 2020 have at least one of the following histologic characteristics: DMI, high-grade tumor, non-endometrioid histological subtype, LVSI, extrauterine spread or nodal involvement [11]. If any of these risk factors were present, more extensive and radical surgery, radiation therapy, or chemotherapy is recommended to improve the long-term prognosis [24]. According to ESMO-ESGO-ESTRO 2020, the recommended treatment for low-risk patients consists of bilateral salpingo-oophorectomy (THBSO) not followed by lymphadenectomy (LA) or adjuvant therapy; nevertheless, a recent study reported that 8.1% of high-risk patients with EC underwent inadequate surgery and 85.4% of low-risk patients with EC underwent LA, resulting in under-treatment or over-treatment [18].

However, clinical and scientific evidence showed that patients in the same risk class have discordant disease progression and that sometimes, low-risk patients may show unexpectedly shorter progression-free survival (PFS) than high-risk patients [5]. This highlights the need for new predictive tissue markers that can integrate the preoperative risk classification, these were identified in molecular mutations underlying the molecular classification proposed by the TGCA and consistent with four distinct subgroups: POLEmut—related to excellent prognosis; microsatellite instability (MSI) status; a “copy-number high” subgroup with p53 mutations—related to the unfavorable course of the disease—and the “copy-number low” subgroup [7].

Molecular preoperative assessment may be used not only for risk stratification but also for tailored surgical decisions, adjuvant therapy and individualized follow-up. An innovative surgical approach may affect the performance of lymphadenectomy in low-risk patients (endometrioid, FIGO stage Ia and G1/2 tumors) if carriers of a molecular mutation linked to unfavorable prognosis or may influence the surgery radicality (systematic lymphonodectomy, omentectomy or multivisceral debulking) [5]. In addition, risk classification may guide the choice of adjuvant therapy; in fact, poly-ADP-ribose polymerase (PARP) inhibitors were shown to be very effective in patients carrying a p53 mutation and MMRd subtype [5]. In addition, correct preoperative identification of tumor aggressiveness allows proper prioritization in terms of surgical urgency to high-risk patients, optimizing the organization of surgical departments.

To the best of our knowledge, this is the first study that proposes a predictive radio-genomics model for low-risk EC patients according to the last European Consensus Conference ESMO/ESGO/ESP 2020 molecular classification. At the current status, not all parameters required for complete risk stratification are obtainable preoperatively; below we report the main limitations noted in clinical practice (LVSI, DM, histological subtype and grading). Firstly, myometrial infiltration (DMI) is a determining factor for staging assessment, in fact, an infiltration >50% of myometrial wall upstaged EC from IA to IB stage and from the low-risk class to intermediate risk class EC. As is well known, MRI is considered the gold standard for the locoregional staging of EC and in particular provides a sensitivity and specificity respectively ranging from 81% to 90% and from 82% to 89% for DMI assessment [25,26]. Nevertheless, myometrial infiltration is highly operator-dependent and affected by wide interobserver variability [27]. Moreover, MDI assessment is even more challenging if the tumor involves uterine horns, if the tumor is overdistended in the uterus wall resulting in thinned myometrium or if it is present in a leiomyomatous uterus characterized by the heterogenous signal intensity of the myometrium [28]. Therefore, in the last few years, some studies proposed a radiomic predictive model to evaluate DMI; firstly, Ueno et al. and Ytre-Haugeet developed a radiomic model based on single slice ROI providing an accuracy, respectively, of 81% and 78% in the identification of DM [19,29]. Recently, Arnaldo Stanzione et al. proposed a radiomic-ML model tested in both a cross-validation and external validation dataset, which provide an overall accuracy of 91% in the detection of DMI on T2-w images after whole-tumor segmentation [26].

The preoperative biopsy samples allow to establish the histologic type and the grade but, as Phelippeau et al. report in their study, there is often discordance between preoperative tissue results and the final surgical examination, probably because of the small amount of tissue that can be examined during the biopsy. This finding was particularly relevant for tumor grading preoperative assessment; Phelippeau et al. reported that the grade was underestimated in up to 29% of tumors [30]. This condition was studied by Yoshiko et al., who related radiomic MRI features to high grade, developing a predictive model with an AUC, sensitivity, specificity, accuracy, positive and negative predictive values of 0.83, 81.0%, 76.8%, 78.1%, 60.7%, and 90.1%, respectively [19].

In recent years, several authors have included all these critical issues in a single radiomic model that included all the risk factors considered for risk stratification of patients with EC. Below we present the results of some radiomic models based on the new non-molecular risk classification.

Recently, Bi Cong Yan et al. developed a whole tumor MRI-radiomic nomogram model combining texture features extracted from T2WI and ADC with clinical-histological data [18]. This study included a large number of patients and showed a high-performance accuracy (AUC of 0.896) in predicting high-risk patients according to the non-molecular risk classification 2020. Similarly, Yan et al. developed a highly predictive radiomic model (AUC values were 0.935 for the training set, 0.909 and 0.885 for validation sets 1 and 2) helping radiologists to improve the assessments of pelvic lymph node metastasis (PLNM) in endometrial cancer (EC) preoperatively [31].

Considering the increasing impact of molecular analysis on the future perspective of personalized therapy, in our retrospective study, we revied all histologic samples in order to complete immunohistochemical stains requested to assess molecular classification of neoplasm: POLE-mutant EC, MMR (mismatch repair)-deficient EC, p53 mutant EC, NSMP (no specific molecular profile)-EC, (sec. WHO classification of Female Genital Tumours, 2019). Therefore, we developed an MRI-based whole-tumor predictive model which distinguished between low and high/intermediate risk according to the ESGO/ESTRO/ESP 2020 molecular classification. Our radio-genomic model was based on Coarseness features (NGTDM) extracted from T2w and Difference entropy (GLCM) extracted from ADC and showed an AUC of 0.75 and 0.74, respectively, for training and testing. Both features selected for our model showed how high intra-tumoral heterogeneity is related to tumor aggressiveness; in fact, higher value Coarseness features (NGTDM) were found in the low-risk class, which is indicative of a lower rate of spatial change and a more locally uniform texture; while entropy (GLCM) reflects the measurement of the randomness/variability in neighborhood intensity values showing higher results in high-risk patients.

Features extracted from both sequences were used in our model; this demonstrated that both T2WI and ADC maps provide significant information about the microstructural composition of the tumor.

Moreover, our model used the same radiomic feature extracted from the ADC map of the predictive model of Ytre-Hauge et al. Therefore, the selection of the same texture feature in different predictive models for high-risk endometrial carcinoma shows concordant results and supports the real link between this radiomic feature and tumor microstructure [29].

Our study has some limitations, primarily the small sample of patients selected because of our strict inclusion criteria and the anatomopathological review of tissues required to evaluate the newly introduced molecular classification. The small group of patients also limits the number of radiomic features considered in the radio-genomic model in order to avoid model overfitting. Our training set for risk stratification was slightly off-balance in favor of high/intermediate risk class between the training and validation set; therefore, we prefer a K-fold cross-validation approach to train and validate the selected model. The retrospective nature of the study, even if the MRI was conducted with a standardized protocol, could bring imaging data inhomogeneity. In addition, we manually performed whole tumor segmentations instead of using automatic/semiautomatic segmentation and this could expose bias related to the subjectivity of evaluation; even if segmentation features dependency was explored. Finally, we included in our model only T2 DWI and ADC map images, while we did not investigate T1 post-contrast images, which may include other significative data.

## 5. Conclusions

In our study, MRI-based whole-tumor radiomic and radio-genomic analysis produced a medium-to-high diagnostic performance to discriminate low-risk EC from other risk classes and the presence of LVSI. The application of predictive models to clinical practice would lead to significant advantages in the preoperative selection of personalized therapy and a reduction in the time/cost of investigations required for subsequent personalized treatment. In recent years, radiomics has taken a central role in scientific research and is proving to be a valuable diagnostic support tool in various fields, including gynecologic oncology, although further investigations are still needed to verify its performance before its effective use in clinical practice.

## Data Availability

The authors are willing to provide the dataset upon reasonable request, the study data were not archived or analyzed publicly.

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
