# Peer review of "MRI- and Histologic-Molecular-Based Radio-Genomics Nomogram for Preoperative Assessment of Risk Classes in Endometrial Cancer"

_cancers, 2022, doi:10.3390/cancers14235881_

Round 1

Reviewer 1 Report

Dear Editor, 

First, I would like to congratulate the authors for their effort on tackling one of the unanswered questions on surgical management of endometrial cancer patients: how can we correctly assess risk stratification before surgery?

I would like to express some doubts and commentaries on the manuscript presented to your journal:

- I would like to ask the authors if the classification undergone for EC molecular classification followed the ProMisE scheme (where MMRd POLEmut patients are classified as MMRd) or the PORTEC-Leuven classification (where MMRd POLEmut patients are classified as POLEmut). In my opinion, as (not so) recent findings emerged from Leon-Castillo (from the Leuven group) that corroborate excellent prognosis for so-known multiple classifiers involving POLEmut, it is not acceptable to classify those patients as MMRd. Please specify the algorithm used to classify patients.

- Moreover, along the introduction section, there are some specifications that need to be clarified. The authors state that the ESMO/ESGO/ESTRO Conference 2016 established 4 groups that took into account LVSI in order to tailor surgery. This is not properly specified, as the groups established by the 2016 Conference refer to prognostic features found postoperatively, and are only meant to guide adjuvant treatment. In the 2016 conference recommendations, "classical" groups for lymph node involvement risk are still used (and specified throughout the discussion of the recommendations). The 2020 guidelines are the ones that do include postoperative factors to classify the patients for both surgery and adjuvant treatment, therefore including LVSI as a factor unknown preoperatively, but necessary to tailor patients surgery. Besides, surgical procedures (lines 90-91) DO include SLN as a staging procedure for all patients (being a mere recommendation or a mandatory procedure to perform depending on the risk of the patient). 

- In addition to my first comment, I find that molecular classification is poorly explained, with several errors that can be spotted through lines 114-122. 

- I feel that the role of TV-US performed by an expert has no recognition through the mansucript, and it is one of the most valuable techniques that we, gynecologist oncologists, have to establish myometrial invasion. Please consider TV-US in the introduction. 

- Standard surgical procedure for EC is not radical hysterectomy type A, that consists on excising 0.5 mm of parametrial tissue around the uterus. This is not a recommendation nor a suggestion in any of the current guidelines, as radical hysterectomy has not a prognostic benefit compared to standard TH in initial-stage EC (line 186). 

- I cannot find any specifications on the surgery performed of the 64 included patients. Please change the Italian word "linfonodi" to "Lymph node status" on Table 4.

- Patient characteristics (lines 325-342) should be summarized and grouped. 

- I was not able to find what is considered as "low" and "high" risk according to the classification performed. On other words, I do not know if "low risk" patients are the same as the ones that would fall under the "low risk" 2020 classification. Therefore, are intermediate risk patients considered as high risk? Only high/low risk patients were taken into account? Is the classification performed considering molecular factors? Please explain. 

Author Response

Reviewer 1:

Thank you for your appreciated revisions and corrections.

Responses to your comments are listed below.

  1. Thank you for the comments. We used ProMisE algorithm; to clarify this we changed the caption of Table 1 and from line 243 to 247 of Pathological analysis. We add comments “No patients had tumors that demonstrated more than one molecular features” in Results-study population line 371 and 382.

  1. We accepted the revision, and we change the introduction; we added “sentinel lymph node (SLN) dissection” in the text.

  1. We accepted the revision, and we change the introduction in line 108 and 126;

  1. We agree that TVS ultrasound performed by gynecologists has a crucial role in the diagnostic pathway of EC, however, we did not focus on the diagnostic role of ultrasound because our goal was to show innovative diagnostic tools extracted from MRI that could provide useful clinical implications. However, in view of its important role, we mentioned it TVS in the introduction.

  1. Our validation of the current classification derives from the contents of these two references:

         Cibula D, Abu-Rustum NR, Benedetti-Panici P, Köhler C, Raspagliesi F, Querleu D, Morrow CP. New classification system of radical hysterectomy: emphasis on a three-dimensional anatomic template for parametrial resection. Gynecol Oncol. 2011 Aug;122(2):264-8. doi: 10.1016/j.ygyno.2011.04.029. Epub 2011 May 17. PMID: 21592548.

    2.     Querleu D, Morrow CP. Classification of radical hysterectomy. Lancet Oncol. 2008 Mar;9(3):297-303. doi: 10.1016/S1470-2045(08)70074-3. PMID: 18308255.

    Hysterectomy type A:  corresponds to the extrafascial hysterectomy, which guarantees full removal of the pericervical tissue up to the attachment of the vaginal fornices.
    Ureteral dissection: the ureter does not need to be unroofed.
    Parametria: this type does not allow for the resection of the ventral or lateral parametria, it does not include resection of the dorsal parametria. The hypogastric plexus, therefore, remains fully preserved. [1,2]

     In order to avoid misunderstanding, we have corrected the manuscript with these clarifications.

  1. Thank you for the comments we modified the manuscript from 204 to 220. We corrected word "linfonodi" to "Lymph node status".

  1. Thank you for the comments, All information of Group1 ad Group 2 has been summarized in Table 4.

  1. Thank you for the comments we modified the manuscript adding a paragraph from line 198 to 203.

Reviewer 2 Report

This is a well written and well researched manuscript that will be of interest to medical professionals involved in gynecology oncology patient care.  This method of imaging provides additional information that can guide patient surgical management.  As you have noted molecular classification for endometrial carcinoma is currently used for risk stratification and adjuvant therapy.

I only have a few suggestions for your consideration.

Line 519: "At current status, not all parameters required for complete risk stratification are obtainable preoperatively; below we report 520
the main limitations noted in clinical practice (LVSI, DM, histological subtype ad grading)."  Could the sentence read -
At current status, not all parameters required for complete risk stratification are obtainable preoperatively; below we report the main limitations noted in clinical practice (LVSI, DM, histological subtype and grading).

Line 528:  "Moreover, MDI assessment is even more challenging 528
if tumor involves uterine horns, if the tumor overdistended the uterus wall resulting in thinned myometrium or if its present a fibromatous uterus characterized by heterogenous signal intensity fo myometrium [28].  Could the sentence read -
Moreover, MDI assessment is even more challenging if tumor involves uterine horns, if the tumor overdistended the uterus wall resulting in thinned myometrium or if its present in a leiomyomatous uterus characterized by heterogenous
signal intensity of the myometrium [28].

Author Response

Reviewer 2:

Thank you for your appreciated corrections, we accepted the revisions and modified the manuscript at line 564-567 and line 573-576.

Round 2

Reviewer 1 Report

Dear Editor, 

I find that the corrections made by the team match the required queries presented earlier.